# Maternal Immunization: Current Evidence, Progress, and Challenges

**DOI:** 10.3390/vaccines13050450

**Published:** 2025-04-24

**Authors:** Veronica Santilli, Mayla Sgrulletti, Giorgio Costagliola, Alessandra Beni, Maria Felicia Mastrototaro, Davide Montin, Caterina Rizzo, Baldassarre Martire, Michele Miraglia del Giudice, Viviana Moschese

**Affiliations:** 1Research Unit of Clinical Immunology and Vaccinology, Academic Department of Pediatrics (DPUO), IRCCS Bambino Gesù Children’s Hospital, 00165 Rome, Italy; 2Pediatric Immunopathology and Allergology Unit, Policlinico Tor Vergata, University of Rome Tor Vergata, 00133 Rome, Italy; maylasg@gmail.com (M.S.); moschese@med.uniroma2.it (V.M.); 3Section of Pediatric Hematologyand Oncology, Azienda Ospedaliero Universitaria Pisana, 56126 Pisa, Italy; giorgio.costagliola@hotmail.com; 4Department of Clinical and Experimental Medicine, University of Pisa, 56126 Pisa, Italy; alessandrabeni95@gmail.com; 5Pediatrics and Neonatology Unit, Maternal-Infant Department, “Monsignor A.R. Dimiccoli” Hospital, 70051 Barletta, Italy; liciamastrototaro@gmail.com (M.F.M.); baldo.martire@gmail.com (B.M.); 6Division of Pediatric Immunology and Rheumatology, “Regina Margherita” Children Hospital, 10126 Turin, Italy; davide.montin@gmail.com; 7Department of Translational Research and New Technologies in Medicine and Surgery, University of Pisa, 56126 Pisa, Italy; caterina.rizzo@unipi.it; 8Department of Woman, Child and General and Specialized Surgery, University of Campania “Luigi Vanvitelli”, 81100 Naples, Italy; michele.miragliadelgiudice@unicampania.it

**Keywords:** immunization, maternal, influenza, pertussis, hepatitis B, COVID-19, pregnancy, immunology

## Abstract

Maternal immunization is a key strategy for protecting pregnant individuals and newborns from infectious diseases. This review examines the mechanisms and benefits of maternal immunization, with a focus on transplacental IgG transfer and immune system interactions. We provide an overview of current recommendations and the safety and efficacy profiles of maternal vaccines, including influenza, tetanus–diphtheria–acellular pertussis (Tdap), respiratory syncytial virus (RSV), COVID-19, and hepatitis B. Additionally, we analyze the barriers to maternal immunization, such as misinformation, vaccine hesitancy, and disparities in healthcare access, while exploring potential strategies to overcome these challenges through targeted educational initiatives, improved provider communication, and policy-driven interventions aimed at increasing vaccine confidence and accessibility. Finally, this review highlights recent innovations and future directions in maternal immunization, including emerging vaccines for Group B Streptococcus and cytomegalovirus. Expanding immunization programs and advancing research on maternal–fetal immunity are essential to optimizing vaccination strategies, improving public health outcomes, and reducing the global burden of infectious diseases.

## 1. Introduction

Maternal immunization represents a cornerstone in the prevention of infectious diseases during pregnancy with the safeguard of both maternal and neonatal health. Pregnant women face severe risks from infections due to immunological changes during gestation, while neonates possess immature immune systems, making them more susceptible to severe infections compared to the general population. Vaccination during pregnancy has thus emerged as a pivotal strategy to address these vulnerabilities. Maternal immunization is one of the most cost-effective and impactful public health interventions, providing dual protection for both the mother and the newborn. By reducing the burden of preventable infectious diseases, vaccination in pregnancy not only safeguards maternal and neonatal health but also decreases healthcare costs, hospitalizations, and long-term complications. Investing in maternal immunization translates into healthier pregnancies, stronger neonatal immunity, and a profound impact on public health.

Over the past decades, substantial progress has been made in understanding the safety and effectiveness of vaccines administered during pregnancy. A systematic review and meta-analysis encompassing randomized controlled trials (RCTs) assessed vaccines such as influenza, tetanus–diphtheria–acellular pertussis (Tdap), and respiratory syncytial virus (RSV). The findings indicated that these vaccines effectively prevent diseases in both mothers and infants, with no significant increase in adverse pregnancy outcomes [1].

Despite the proven safety and benefits, vaccination coverage in pregnancy remains suboptimal in many countries. Several barriers, including misinformation, lack of healthcare provider recommendations, and vaccine hesitancy, contribute to these gaps.

This review aims to provide a comprehensive update on maternal immunization, delving into the following three key aspects: (1) international recommendations and guidelines; (2) data on efficacy, effectiveness, and safety; and (3) recent highlights from the scientific literature and future prospective research on vaccines during pregnancy (Figure 1).

## 2. Mechanisms and Benefits of Maternal Immunization

The fetal and newborn immune system presents some developmental peculiarities. Indeed, it is shifted towards a T-helper 2 (Th2) immune response, thus being ineffective in the defense against intracellular pathogens [2]. Moreover, the ability to mount a T-independent antibody response (i.e., against polysaccharides such as bacterial capsular antigens) is impaired in the first two years of life. This significantly affects the infectious risk in the newborn, the impact of passive maternal antibody transfer, and, therefore, the importance of maternal immunization.

Therefore, the role of the fetal–maternal interface is pivotal for newborn immune homeostasis and protection against a wide range of infectious diseases [2,3]. This interface has historically been mostly identified with the passive transfer of adaptive immunity. However, its role in both innate and adaptative immune responses, as well as in the development of active immunity in the newborn, is now well recognized. Maternal innate immunity plays a major role in preventing the vertical transmission of pathogens and acts through cellular components (decidual innate immune cells, placental-associated macrophages and monocytes, fetal placental macrophages) and cytokines [4].

The functions of adaptive immunity, particularly the passive maternal antibody and lymphocyte transfer, are multiple and are directly connected with the effects of maternal immunization [3], as further discussed. Indeed, they allow the transfer of antigen-specific immunity, thus having a fundamental role in the protection against vaccine-preventable diseases.

Among immunoglobulin, IgG is the most efficiently transferred isotype, with IgG1 being the predominant subclass, particularly in responses to protein antigens [5]. The delivery of IgG relies on several pathways, of which the transcytosis mediated by the neonatal Fc receptors expressed by syncytiotrophoblast is the most relevant [3,6,7]. The active transfer of IgG takes place during the second and third trimesters, and it is significantly enhanced after 32 weeks of gestation. Therefore, immunoglobulin levels differ significantly between term and preterm infants [8]. Other factors that influence the effectiveness of antibody transfer include placental integrity, chronic maternal infections, and maternal immunoglobulin levels [9]. The half-life of maternal antibodies in the infant’s blood is variable, ranging from 28 to 35 days, as demonstrated in a recent study by Ogutiet al [10]. The factors that influence maternal antibody half-life are partly unexplored and involve a specific IgG subclass and glycosylation profile [3]. However, it has been demonstrated that, after maternal vaccinations, specific antibodies can persist over 6 months and that the timing of the vaccination significantly affects the antibody concentration and persistence [11,12]. As the half-life of maternally derived antibodies and its determinants can be a relevant factor in defining newborn infectious risk, further studies are needed on this aspect. Moreover, antibodies can be passively delivered through lactation. In colostrum and breastmilk, IgA represents the dominating antibody isotype, with higher concentrations in colostrum. On the other hand, the concentration of IgG is remarkably lower compared to peripheral blood, as antibodies are transferred to the mammalian alveoli with isotype-dependent mechanisms [13,14].

The above-mentioned immune interface provides a strong rationale to elucidate the mechanisms underlying the effectiveness of maternal immunization in the prevention of fetal and neonatal infections. Indeed, maternal immunization significantly increases the concentration of pathogen-specific antibodies in maternal circulation, enhancing the antibody transfer [15,16].

Notably, beyond the direct protection against infectious diseases, maternal antibodies have complex interactions with the fetal and newborn immune system development. They can reduce the differentiation of memory B cells and plasma cells, and they can mask or remove antigens with potential interference with the generation of an autonomous immune response by the newborn [14,17]. However, this aspect remains partially unclear, and several studies have not demonstrated a reduction of newborn autochthonous immune response after maternal immunization [17]. Additionally, according to the hypothesis of the original antigenic sin, memory B cells are more prone to respond to previously encountered epitopes in the presence of a variable degree of similarities between antigens [18]. This mechanism could potentially influence the future response to some pathogens with high epitope variability (i.e., influenza) when their antigens have been encountered during gestation. Therefore, this aspect must be considered in the analysis of the dynamic fetal–maternal immunological interface.

Interface also provides a transfer of lymphocytes. Indeed, maternal microchimerism can be detected in cord blood and in various tissues [19]. It mostly involves T lymphocytes, with a prevalence of memory T cells [3]. Moreover, breast milk contains a high proportion of B cells and effector T cells [20], with a remarkably different lymphocyte distribution compared to maternal peripheral blood. Therefore, it can be hypothesized that cells are transferred with a selective modality to allow a neonatal rapid defense against pathogens [20]. The effective transfer of pathogen-specific lymphocytes has been recently demonstrated in a study by Armistead et al., where maternal SARS-CoV-2-specific memory T lymphocytes were identified in newborns from mothers vaccinated during lactation [21].

Finally, the phenomenon of transfer of trained innate immunity (TRIM) to offspring might represent an additional benefit of maternal immunization. Indeed, maternally derived cytokines and exposure to specific antigens can affect the prenatal innate immune function, as recently reported [22]. Further elucidation of the TRIM mechanisms in maternal immunity could provide new insights into how prenatal immune priming can influence neonatal immune development. This emerging field may help refine maternal vaccination strategies and optimize early life immunity [22].

## 3. Current Recommendations and Safety Profile of Maternal Vaccines

### 3.1. Influenza Vaccine

Pregnant and post-partum women are at higher risk of severe illness and complications from influenza, particularly during the second and third trimester [23]. Indeed, the Advisory Committee on Immunization Practices (ACIP) and the American College of Obstetricians and Gynecologists recommend that persons who are pregnant or who might be pregnant or post-partum during the influenza season receive an inactivated influenza vaccine (IIV) [24,25].

The live attenuated influenza vaccine is contraindicated during pregnancy for the potential, albeit low, to cause infection in the mother and in developing fetus [24].

Influenza vaccination is considered safe and advisable at any point during pregnancy, regardless of the trimester, and it may be given either prior to or throughout the influenza season. In countries where the vaccine is available, early immunization in July or August may be recommended for women in their third trimester, as it can provide passive protection to newborns during the initial months of life, when they are too young to be vaccinated themselves [26]. Conversely, for those in their first or second trimester during these months, it may be preferable to delay vaccination until September or October, unless there is a risk that immunization might not occur later. Notably, research by W. Cuningham et al. indicates that women vaccinated later in pregnancy exhibit significantly higher hemagglutination inhibition (HI) titers, both in maternal and cord blood, compared to those who received the vaccine earlier [27].

#### 3.1.1. Safety

Extensive evidence supports flu vaccine safety, with no association with adverse pregnancy outcomes such as congenital anomalies or miscarriage [28].

Experience with the use of recently approved influenza vaccines, such as cell culture-based (ccIIV) and recombinant vaccines (RIV), during pregnancy is more limited compared to older formulations. For ccIIV, analyses of data from the Vaccine Adverse Event Reporting System (VAERS) spanning 2013–2020 [29] and a prospective cohort study conducted from 2017–2020 [30] identified no unexpected safety concerns in pregnant individuals compared to other influenza vaccines used in pregnancy. Additionally, recent studies, including a randomized controlled trial (RCT) performed at Clinical Immunization Safety Assessment (CISA) project sites, which evaluated the safety of the recombinant quadrivalent influenza vaccine (RIV4) versus IIV4 in 382 pregnant individuals, provided further evidence supporting the safety of RIV4 during pregnancy [31,32]. Further, no adverse perinatal outcomes have been observed in women vaccinated over close and successive pregnancies, irrespective of vaccine type and interval between pregnancies [33].

#### 3.1.2. Efficacy and Effectiveness

Maternal flu immunization is a critical strategy for protecting both maternal and neonatal health. Indeed, vaccination reduces the risk of influenza-related hospitalization in pregnant individuals by 50–70%. Transplacental antibody transfer provides up to 6 months of passive protection for infants, significantly lowering the incidence of severe respiratory illness [23]. Studies, including retrospective analyses and randomized controlled trials (RCTs), demonstrate that vaccination significantly reduces severe influenza outcomes, such as hospitalization, and prevents adverse birth outcomes. A Centers for Disease Control and Prevention (CDC) study (2022) linked influenza vaccination to a lower risk of fetal death, preterm birth, and low birth weight. Researchers retrospectively analyzed data from New Zealand, covering 16 years, and confirmed that maternal vaccination decreases influenza-related complications during pregnancy [34].

### 3.2. Tetanus, Diphtheria, and Acellular Pertussis (Tdap) Vaccine

The Tdap vaccine during pregnancy is globally recommended to protect infants aged <2 months, the group most vulnerable to severe pertussis outcomes, including hospitalizations and mortality [35]. Tdap vaccination during pregnancy is even more relevant in developing countries where it can prevent neonatal tetanus, a life-threatening condition caused by the toxin produced by Clostridium tetani, which enters the body through contaminated wounds, often during unclean delivery practices or umbilical cord cutting in unhygienic conditions. In newborns, the mortality rate is exceptionally high, particularly in resource-limited settings with limited access to timely and adequate medical care. International guidelines, endorsed by the World Health Organization (WHO) and implemented in many countries, advocate that one dose of the Tdap vaccine be administered between 27 and 36 weeks of gestation. This timing maximizes maternal antibody transfer to the fetus, conferring passive immunity that bridges the critical early months before infants receive their own vaccinations.

To ensure protection against maternal and neonatal tetanus, pregnant women who have never been vaccinated against tetanus should receive three doses scheduled at 0 weeks, 4 weeks, and 6 through 12 months. For previously vaccinated women, one DTap dose should preferably be administered between 27 and 36 weeks of gestation [36].

#### 3.2.1. Safety

The vaccine is well-tolerated and safe for pregnant women, with no significant increase in adverse pregnancy outcomes such as preterm birth, stillbirth, or low birth weight [37].

Since the acellular pertussis vaccine is only available in combination with tetanus and diphtheria toxoids, some concerns have been raised on its administration shortly after a TT or Td vaccine that might increase the risk of adverse reactions. Initial U.S. guidelines recommended a 2-year interval between TT or Td and Tdap for postpartum women [38], but this requirement was removed in 2011–2012 following further evidence [39]. Despite lingering concerns about repeated doses of tetanus-containing vaccines, a study by Sukumaran et al. involving over 29,000 pregnant individuals demonstrated no significant differences in adverse events or birth outcomes based on booster interval [40].

Furthermore, the co-administration of Tdap with the influenza vaccine during pregnancy showed no risk increase. While ongoing monitoring is necessary, current data strongly support the safety of the Tdap vaccination in pregnancy, even with repeated doses at short intervals [41].

#### 3.2.2. Efficacy and Effectiveness

Several real-world serology studies on recombinant pertussis vaccination in pregnancy have reported high anti-PT antibody levels against vaccine antigens—pertussis toxin (PT), pertactin (PRN), and filamentous hemagglutinin (FHA)—in cord samples from women vaccinated with the Tdap vaccine [42,43]. Interestingly, an earlier Tdap immunization, between 27 and 30 weeks gestation, may be associated with higher cord blood pertussis antibody titers [44]. However, the optimal timing for Tdap administration during pregnancy remains debated, with studies suggesting different recommendations, but early vaccination is more likely to be recommended to protect preterm neonates and to extend vaccine opportunities.

The placental antibody transition results in an efficacy of maternal Tdap vaccination in preventing infant pertussis of over 90% for severe cases in the first three months of life, as confirmed by several large cohort studies addressing the effectiveness and clinical relevance of Tdap maternal immunization [44,45,46].

The potential interference between maternal antibodies and the infant’s response to routine pertussis vaccinations at two to three months of age has been investigated. Infants born to vaccinated mothers showed a lower immune response to primary pertussis vaccination [47], with reduced antibody concentrations to several pertussis antigens and lower antibody avidity [48]. Studies have reported that, even in the presence of dampened antibody levels, maternal antibodies do not substantially hinder infant T-cell priming following primary vaccination. However, a system vaccinology approach is required to unravel the possible interaction with the other components of the infant’s developing immune system, including T-cell-mediated immunity, innate responses, and connections with the upper respiratory microbiota, to explore blunting impacts on long-term protection against pertussis [49]. This “blunting” effect could also affect the immune response to other vaccines, such as polio and pneumococcal conjugate vaccines, which use diphtheria or tetanus toxins as carrier proteins [50]. However, extensive research indicates that, while maternal antibodies can slightly reduce the immunogenicity of infant vaccines, they do not compromise clinical protection. This transient interference is outweighed by the immediate and substantial protection maternal antibodies provide to newborns during their most vulnerable period [51].

The recent increase in whooping cough in babies aged <3 months may be due to several factors. Pertussis epidemics naturally occur cyclically every 3–5 years, even with stable vaccine coverage. Immunity from an early life pertussis vaccine wanes over time, and most adults lack protective antibodies, allowing for infection and disease transmission. Acellular pertussis vaccines prevent disease but do not stop infection or its spread, favoring periodic outbreaks that severely affect unprotected infants. Further, the COVID-19 pandemic likely exacerbated infection susceptibility due to significant declines in DTaP vaccine administration observed in 2020–2022. In addition, the emergence of Bordetella pertussis strains with genetic divergence, such as pertactin deficiency and increased toxin production, might compromise vaccine efficacy and drive outbreaks. Low maternal vaccination coverage also renders infants more vulnerable during the critical first few weeks of life. Thus, higher maternal vaccination rates could significantly reduce infant pertussis cases, hospitalizations, and mortality [52].

Studies indicate that immunizing pregnant women with tetanus toxoid (TT) is highly effective in preventing neonatal tetanus (NNT) [53]. Early 1960s clinical trials demonstrated the high efficacy of tetanus toxoid in preventing neonatal tetanus in high-incidence regions [54]. A 94% reduction in neonatal tetanus mortality was observed with three doses of fluid tetanus toxoid, while no deaths occurred among infants of mothers immunized with two or three doses of aluminum-adsorbed toxoid within five years [55]. Later observational and case-controlled studies confirmed vaccine effectiveness ≥ 80% [56,57].

### 3.3. COVID-19 Vaccine

Pregnant individuals are at higher risk of severe COVID-19 outcomes. In fact, they are three times more likely to need ICU care, 2.4 times more likely to need extracorporeal membrane oxygenation (ECMO), and 1.7 times more likely to die [58]. Infection during pregnancy is also associated with adverse fetal outcomes, such as preterm birth and stillbirth. Since COVID-19 mRNA vaccines are currently approved only for individuals aged ≥6 months [59], maternal vaccination could help to prevent COVID-19-related hospitalization in infants too young to be vaccinated, i.e., during the first 3 months of life. mRNA-based COVID-19 vaccines (e.g., Pfizer-BioNTech and Moderna) are currently recommended at any trimester [60]. Booster doses are recommended to sustain immunity, particularly during high-risk periods of viral transmission.

#### 3.3.1. Safety

Data from global surveillance systems and clinical studies demonstrate the safety of COVID-19 vaccines in pregnancy. A recent large population-based study from Sweden and Norway (including 94,303 infants exposed to the COVID-19 vaccination during pregnancy and 102,167 control infants) found no increase in adverse neonatal events in infants born to mothers vaccinated against COVID-19 during pregnancy. In another study, exposure to the COVID-19 vaccination during pregnancy was associated with reduced rates of non-traumatic intracranial hemorrhage, hypoxic-ischemic encephalopathy, and neonatal mortality [61,62]. Mild side effects such as injection site pain, fatigue, and fever are common but transient. While current evidence supports the short-term safety and efficacy of maternal immunization with mRNA COVID-19 vaccines, particularly in reducing the risk of severe maternal disease and neonatal infection, it is important to acknowledge that long-term data—especially regarding outcomes beyond the first few years of life—are still being collected. Caution and continued surveillance remain essential, as with all medical interventions during pregnancy.

#### 3.3.2. Efficacy and Effectiveness

COVID-19 vaccination reduces the risk of severe disease in pregnant individuals by over 90% [63]. Regarding the immunogenicity of the vaccine, it induces significant seroprotective antibody titers and the activation of antigen-specific cellular immunity. In particular, mRNA-based vaccines elicit a strong IgG response capable of crossing the placenta, conferring passive immunity to the fetus [64]. A recent systematic review and meta-analysis of controlled and randomized clinical trials showed that vaccinated pregnant women exhibit a significantly higher amount of postpartum antibody titers compared to that observed in both unvaccinated mothers and mothers who have recently recovered from a SARS-CoV-2 infection [65]. Additionally, maternal vaccination benefits the infant with reduced risk of infant infection and hospitalization. A study conducted by the CDC reported that the effectiveness of maternal vaccination during pregnancy in preventing COVID-19 hospitalization in infants younger than 6 months was 61% [66].

### 3.4. Hepatitis B Vaccine

The prevention of mother-to-child transmission (MTCT) of hepatitis B virus (HBV) is a global public health priority. The hepatitis B vaccine is important for preventing mother-to-child transmission of the hepatitis B virus. However, the role of the hepatitis B vaccine as part of maternal immunization remains unclear. Current strategies focus on screening programs of pregnant women and, in children born to seropositive mothers, on the administration of hepatitis B immunoglobulin (HBIG), of the first dose of the hepatitis B vaccine to newborns within 12 h of birth, and with completion of the vaccine series by six months of age [67]. Vaccination of an HBV-infected mother during pregnancy offers the advantage of maternal viremia decline with risk of intrauterine and peripartum HBV transmission. This approach targets the primary source of infection—maternal blood and body fluids—before the infant is exposed during delivery. In contrast, neonatal vaccination primarily aims to prevent chronic HBV infection after exposure has occurred. Maternal vaccination could synergize with neonatal immunization and HBIG administration to maximize protection, especially in high-risk cases, potentially enhancing immune priming and reducing transmission rates more effectively. A pivotal systematic review by Chen et al. observed that maternal vaccination in HBsAg-positive pregnant women, when combined with neonatal immunoglobulin administration at birth (HBIG), results in a marked reduction in perinatal transmission rates, with no reported adverse outcomes for mothers or infants [68].

Furthermore, novel delivery approaches, such as mRNA-based HBV vaccines, are currently under investigation, promising enhanced immunogenicity and safety profiles [69]. These advances have prompted discussions about revising current HBV management guidelines to recommend routine vaccination for HBV-positive pregnant women, especially in high-prevalence regions. A recent cost-effectiveness analysis by Prabhu M, et al. (2022) [70] on universal hepatitis B antibody screening and vaccination in pregnancy further highlighted the economic and clinical benefits of adopting a universal vaccination strategy. The study demonstrated that such an approach could significantly reduce healthcare costs and improve neonatal outcomes by preventing undiagnosed HBV cases, particularly in low-resource settings [70]. Future research should focus on long-term outcomes and cost-effectiveness, optimizing vaccination timing for maximum benefit.

### 3.5. Respiratory Syncytial Virus (RSV) Vaccine

The respiratory syncytial virus (RSV) is a leading pathogen responsible for bronchiolitis and viral pneumonia in infants. The highest risk period occurs within the first three months of life, during which hospitalization rates peak, with a mortality rate of 1–3% among hospitalized cases. While factors such as prematurity, chronic pulmonary or cardiac disorders, and immunodeficiencies significantly elevate the likelihood of severe outcomes; even otherwise healthy infants under six months are at notable risk for RSV-associated complications [71]. Preventive measures play a pivotal role in mitigating the impact of RSV. A major breakthrough in maternal immunization is the development of the RSV vaccine. On 21 August 2023, the FDA approved Abrysvo for use in pregnant individuals between 32 and 36 weeks of gestation [72]. Abrysvo incorporates two recombinant stabilized RSV prefusion F antigens from the RSV-A and RSV-B subgroups. The prefusion F protein is the primary target of neutralizing antibodies, which, through transplacental transfer, prevent RSV infection in newborns, protecting infants from birth up to six months [73].

#### 3.5.1. Safety

The safety of Abrysvo was evaluated in a large clinical study where approximately 3600 pregnant participants received a single dose of Abrysvo compared to pregnant controls who received a placebo [74].

The most frequently observed adverse events in Abrysvo-receiving participants included local injection discomfort, headaches, musculoskeletal pain, and nausea. The majority of these effects were of mild to moderate intensity and typically resolved within a few days.

An imbalance in the occurrence of preterm births was observed in the Phase 3 trial among pregnant individuals who received the Abrysvo vaccine, with this imbalance primarily seen in participants from low- to middle-income countries. The FDA was concerned that the observed 1% difference in preterm births (5.7% among Abrysvo recipients and 4.7% among placebo recipients in the phase 3 trial) might represent a true vaccine-associated risk of preterm birth [75].

Given this concern, in order to avoid the potential risk of preterm birth with use of Abrysvo before 32 weeks of gestation, FDA decided to approve vaccine administration in pregnant individuals at 32 through 36 weeks of gestational age. In the phase 3 trial, considering this immunization period, 4.2% in the Abrysvo group and 3.7% in the placebo group were born preterm [75]. Thus, narrowing the time window of maternal immunization avoids the potential risk of preterm birth before 32 weeks of gestational age, i.e., very preterm or extremely preterm births, in the case further studies confirm the true risk [75].

Similarly to the United States, several countries worldwide have authorized Abrysvo at varying gestational age windows. The UK, France, Belgium, Canada, Argentina, and Australia have opted for restricted gestational age windows to reduce the risk of preterm births and of higher neonatal mortality and long-term morbidity [76].

In addition, although not commonly reported, pre-eclampsia occurred in 1.8% of pregnant individuals who received Abrysvo compared to 1.4% of pregnant controls [74].

As of July 2024, four post-marketing investigations have planned or commenced to evaluate the association between vaccine use and the risk of preterm birth and hypertensive pregnancy disorders, including pre-eclampsia [77,78]. Preliminary results on the RSVpreF maternal vaccine showed no risk of preterm birth.

Data on the use of Abrysvo are constantly monitored, and potential side effects carefully evaluated to take any necessary action for protection.

#### 3.5.2. Efficacy and Effectiveness

A Phase 3, randomized, double-blind, multicenter, placebo-controlled study (NCT04424316-C3671008) [74] demonstrated that Abrysvo reduced the risk of RSV-LRTD by 51% in infants born to vaccinated mothers compared to controls. Of the 3495 infants born to vaccinated mothers, 57 developed RSV-LRTD within the first 6 months of life, compared to 117 out of the 3480 infants born to control mothers. The vaccine reduces RSV-related hospitalizations in neonates by approximately 70–80% during the first 90 days of life. It also lowers the severity of RSV-associated respiratory illness in newborns. As RSV is a leading cause of neonatal hospitalization globally, this vaccine is expected to have a substantial public health impact. However, the administration of the monoclonal antibody nirsevimab to infants at high risk of severe RSV disease, including those born to vaccinated mothers, is currently under consideration. The U.S. CDC recommends that infants receive protection against RSV through either maternal vaccination during pregnancy or the direct administration of nirsevimab after birth. However, the CDC advises that both interventions are not typically necessary for most infants [79].

This guidance underscores the importance of evaluating individual risk factors when determining the optimal preventive strategy for RSV in infants.

## 4. Barriers to Maternal Immunization

Despite strong recommendations from the WHO and national health authorities, maternal immunization remains globally underutilized. This is of great concern given the substantial benefits for both mothers and infants. Even in countries with established national immunization programs, vaccination coverage rates among pregnant women remain low. In fact, in Ireland, influenza and COVID-19 vaccine uptake among pregnant women reached 62% and 25%, respectively, while in the UK, Public Health England reported a seasonal influenza vaccine rate of 44%. In Italy, these rates are generally lower, with high regional variability in vaccine coverage rates, ranging from 6% to 19% for influenza, 5% to 61% for pertussis, and approximately 20% for COVID-19 vaccines. In the US, recent estimates indicate coverage rates of 61%, 57%, and 70% for influenza, pertussis, and COVID-19 vaccines, respectively [80,81,82,83]. Moreover, the latest CDC report highlighted that only 33% of eligible pregnant women in the United States received the RSV vaccine during the 2023–2024 season [84].

Lack of perceived infection susceptibility and misinformation about vaccine effectiveness represent two of the main determinants of poor maternal vaccine adherence. Despite the existence of maternal immunization programs, studies have shown significant variability in women’s awareness of these initiatives (25–39%) [85,86]. Many pregnant women may be unaware of their risk of contracting pertussis and passing it on to their newborn, who is particularly vulnerable to severe complications. In the same way, some pregnant and postpartum women may not fully understand their increased risk of severe influenza infection and the effectiveness of the flu vaccine. Perceiving oneself as at risk for a disease and understanding the severity of the illness are crucial factors in motivating vaccine acceptance. In a study of more than 170 pregnant women, concern about contracting influenza was a strong predictor of their intention to get vaccinated [87]. On the other hand, preconceptions and prior vaccination behaviors can negatively influence a woman’s decision to get vaccinated during pregnancy. Indeed, some women may erroneously believe that vaccines are primarily for children and that adults, including pregnant women, do not need them. This misconception can lead to vaccine hesitancy or outright refusal [88,89].

Other primary barriers to maternal immunization are represented by widespread concern regarding the safety of vaccines for both the mother and the fetus and doubts about vaccine efficacy. Several studies have shown that 29% of pregnant women believed vaccines could harm fetal development, while 18% doubted their effectiveness in protecting infants from pertussis in the early months of life [86,90]. Negative personal or family negative with vaccines can generate fears and concerns, making women more reluctant to get vaccinated. These experiences may include adverse reactions or misconceptions about vaccine safety [91]. Recent research has shown that women who perceive the influenza vaccine as unsafe are 86% less likely to accept it during pregnancy [92]. An Italian cross-sectional study involving 1031 women and including 553 pregnant women, identified “fear of vaccine complications” (43%) and “concern about vaccine excipients” (12%) as the primary reasons for refusing influenza and pertussis vaccines during pregnancy [86].

In this view, it is essential to consider the potential consequences of attributing neonatal or congenital outcomes to maternal vaccination without robust scientific evidence. Misattribution may not only lead to unwarranted fear and vaccine hesitancy but can also place undue psychological and legal burdens on families and healthcare providers. The importance of clear, evidence-based communication about the risks and benefits of vaccination during pregnancy cannot be overstated, especially given the complexities of causality and the natural incidence of adverse perinatal outcomes.

A multi-faceted approach combining health education, effective communication, trust-building, and community engagement is crucial to overcome barriers to vaccination during pregnancy and to promote the health of mothers and infants.

Access to adequate prenatal care is a crucial factor affecting maternal vaccination rates, both in high-income countries and in low- and middle-income countries [91]. Studies have consistently shown a positive association between the number of prenatal care visits and vaccine uptake. For instance, research conducted in Ethiopia, Pakistan, Ivory Coast, Sierra Leone, Peru, and Brazil has demonstrated that increased prenatal care visits are linked to higher rates of vaccine acceptance [93,94,95,96,97,98].

However, access to care can be hindered by various factors, including socioeconomic barriers (e.g., poverty, lack of insurance, and limited access to transportation), cultural and social factors, and healthcare system-related factors (e.g., inadequate healthcare infrastructure, insufficient healthcare providers, and long waiting times). Even in high-income countries, access to adequate prenatal care can be a challenge. Studies in the United States and Europe have shown that women with government-funded insurance or inadequate prenatal care are less likely to receive influenza and Tdap vaccinations during pregnancy [99,100,101,102,103,104,105,106]. Expanding access to affordable, high-quality prenatal care, especially in underserved communities, and policy changes (e.g., reimbursement for vaccine administration) could facilitate vaccine uptake.

Certain groups of pregnant women appear to be more susceptible to vaccine hesitancy [91]. In particular, studies have shown that women with lower socioeconomic status, less education, and those belonging to racial or ethnic minority groups are less likely to get vaccinated. This trend is likely influenced by sociocultural and psychological factors affecting vaccine acceptance, as seen in non-pregnant populations [86,88]. In the Italian Vax4globecross-sectional survey, vaccine awareness and immunization compliance in childhood and pregnancy were investigated in 310 parent/child pairs of non-European (non-EU) origin. We found that maternal anti-pertussis, SARS-CoV-2, and flu vaccines were received by 15%, 13%, and 3.5% women, respectively, and that only 10% of women received all three vaccines during pregnancy. Vaccine uptake was particularly low among women belonging to non-EU European States. This lack of vaccination acceptance was mostly linked to poor information on maternal immunization. Women with lower education levels were also more likely to miss out on essential vaccines [107]. Additionally, factors such as limited access to healthcare, language barriers, socioeconomic status, and cultural beliefs can further hinder vaccine uptake among specific groups of women. Some cultures may have traditional healing practices that compete with Western medicine, or there may be mistrust of governmental institutions, including healthcare systems. Distrust in healthcare institutions, fueled by personal negative experiences or misinformation, can undermine confidence in vaccines and discourage vaccination [91]. To address these disparities, it is essential to implement targeted interventions that consider the specific needs and barriers of different racial, ethnic, and linguistic groups. These interventions may include cultural competency training, language access services, and the development of culturally appropriate messaging and educational materials. Collaboration with local communities and religious organizations can help disseminate accurate information and build trust in vaccines.

Physician recommendations were associated with higher vaccination coverage, underscoring the crucial role of healthcare providers in fostering the adoption of preventive measures [108]. A survey of 528 women in Ireland on maternal knowledge, attitudes, and practices regarding RSV vaccination during pregnancy revealed that general practitioners were identified as the preferred source of vaccine guidance and that76% found the vaccine acceptable to protect their infants [109]. It goes without saying that a major reason for low vaccination rates is the lack of recommendation or even discouragement from healthcare providers [91]. Many healthcare providers, including obstetricians, midwives, and practice nurses, express vaccine concerns that often stem from a lack of adequate training and education regarding vaccine safety and efficacy. Indeed, a survey of maternity care providers in Canada revealed that the most common reasons not to recommend the flu vaccine were concerns about potential risks, suboptimal efficacy, and lack of information about recommended vaccines [110]. Despite recognizing the importance of the flu vaccine for pregnant women, many healthcare providers feel ill-prepared to address the topic and would benefit from improved access to evidence-based information. Similarly, a study of healthcare providers in the UK and Australia found that a significant proportion of participants felt they had insufficient training on maternal immunization [111,112].

Addressing these concerns requires comprehensive education and training programs for healthcare providers. By providing accurate and up-to-date information on vaccine safety and efficacy, healthcare providers can dispel misconceptions and encourage confident vaccine uptake. In this scenario, collaborative efforts between healthcare providers, public health officials, and patient advocacy groups can help overcome barriers and promote vaccine acceptance.

## 5. Recent Innovations and Future Directions

Several new vaccines are under development targeting pathogens with a substantial disease burden during pregnancy, including Group B Streptococcus (GBS), respiratory syncytial virus (RSV), cytomegalovirus (CMV), Zika virus, HIV, malaria, and extraintestinal Escherichia coli.

GBS remains a major cause of neonatal sepsis, pneumonia, meningitis, stillbirth, and preterm birth [113]. Current preventive strategies, such as universal maternal screening and intrapartum antibiotic prophylaxis, are effective against early onset disease but are insufficient for late-onset disease and adverse birth outcomes, particularly in resource-limited settings. Vaccine development has progressed from monovalent to trivalent conjugate vaccines targeting serotypes Ia, Ib, III, and, more recently, to a hexavalent glycoconjugate vaccine covering up to 98% of colonizing strains [114]. Early trials demonstrated robust immunogenicity and effective maternal antibody transfer, with modeling suggesting that worldwide GBS maternal vaccination could prevent 127,000 GBS cases and 37,000 infant deaths annually [115].

CMV, a leading cause of congenital infections and sensorineural hearing loss, poses challenges due to limited understanding of fetal immune correlates of protection. Nonetheless, vaccine development is ongoing, exploring platforms such as live-attenuated, vectored, DNA, and mRNA vaccines [116]. Similar efforts target malaria, with vaccines in development focusing on placental-binding proteins to prevent adverse pregnancy outcomes [117]. Beyond specific vaccine candidates, the future of maternal immunization depends on a multifaceted approach encompassing research, policy, and healthcare delivery. Strengthening global immunization platforms is crucial, particularly in low- and middle-income countries where the burden of vaccine-preventable diseases is highest and access to maternal healthcare is often limited. Many of these countries have not yet fully implemented routine maternal vaccinations, such as Tdap and influenza, highlighting the need to establish robust infrastructures capable of integrating new vaccines as they become available.

Key areas for future research include a deeper understanding of the maternal–fetal interface and the immunological mechanisms that govern transplacental antibody transfer. Identifying immune correlates of protection is essential for accelerating vaccine development, especially for pathogens like CMV and GBS, where conducting large-scale efficacy trials may be challenging due to low disease incidence. Additionally, investigating novel vaccine platforms, delivery methods, and immunization schedules can optimize vaccine efficacy and safety across diverse populations and healthcare settings.

Global collaboration among researchers, public health organizations, policymakers, and industry stakeholders is vital to ensure rapid and equitable access to new maternal vaccines. International initiatives focus on establishing frameworks that support evidence-based decision-making, harmonize regulatory standards, and accelerate vaccine approval and distribution. These efforts involve creating networks for sharing data and best practices, which simplify regulatory processes and reduce delays. The overall goal is to ensure that new maternal vaccines can be implemented promptly and safely on a global scale.

## 6. Conclusions

In conclusion, maternal immunization has emerged as a critical strategy for safeguarding both mothers and infants from infectious diseases. Evidence from vaccines against influenza, pertussis, and COVID-19 highlights the benefits of this approach, which works primarily through transplacental IgG transfer and the transfer of memory T lymphocytes during later pregnancy. Emerging candidates for pathogens like Group B Streptococcus, respiratory syncytial virus, and cytomegalovirus offer promising avenues to further enhance neonatal protection. However, despite these advancements, vaccination uptake remains suboptimal in many regions, largely due to socioeconomic disparities, limited access to prenatal care, and ongoing vaccine hesitancy. Addressing these challenges requires a multifaceted approach that includes improved healthcare infrastructure, enhanced provider education, and targeted public health initiatives designed to build trust and dispel misconceptions. Furthermore, continued research into the maternal–fetal interface and immune correlates of protection is essential for refining immunization strategies. Ultimately, integrating maternal immunization into routine prenatal care not only holds the promise of reducing the global burden of infectious diseases but also contributes to the broader goal of health equity and improved public health outcomes for future generations.

## Figures and Tables

**Figure 1 vaccines-13-00450-f001:**
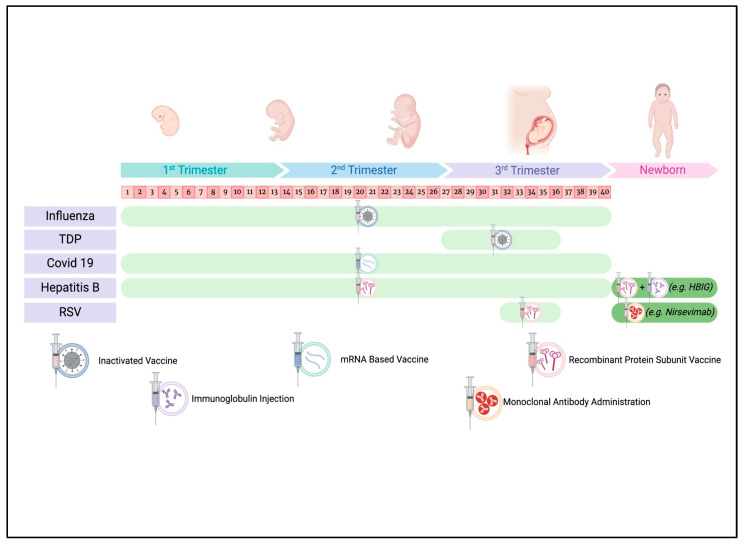
Recommended timeline of preventive measures during pregnancy and in the newborn. Figure created by https://www.biorender.com/ (accessed on 12 December 2024).

## Data Availability

Not applicable.

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
