# Peer review of "Maternal Immunization: Current Evidence, Progress, and Challenges"

_vaccines, 2025, doi:10.3390/vaccines13050450_

Round 1

Reviewer 1 Report

Comments and Suggestions for Authors

This paper is a valuable review on "maternal immunization" and includes “current evidence, progress, and challenges”.

Major Comments

・The hepatitis B vaccine is important for preventing mother-to-child transmission of the hepatitis B virus. However, the role of the hepatitis B vaccine as part of maternal immunization remains unclear.

Minor Comments

  • line 204-205: The lines are misaligned.
  • line 322: Vaccination o an HBV infected mother → Vaccination of HBV-infected mother ?
  • line 482: Vax4globecross-sectional survey → Vax4globe cross-sectional survey ?
  • line 509: health provider91. → health provider [91].

Author Response

We thank you for your  suggestions. We have revised our paper according to your comments.

Comments 1: The hepatitis B vaccine is important for preventing mother-to-child transmission of the hepatitis B virus. However, the role of the hepatitis B vaccine as part of maternal immunization remains unclear.

Response 1: Thank you for pointing this out. We have modified the text of the manuscript according introducing your sentence (Pag.9 line 341-342). 

Comments 2: line 204-205: The lines are misaligned.
Response 2: thank you. We have modified the text. 

Comments 3: line 322: Vaccination o an HBV infected mother → Vaccination of HBV-infected mother ?
Response 3: thank you. We have modified the text. 

Comments 4: line 482: Vax4globecross-sectional survey → Vax4globe cross-sectional survey ?
Response 4: thank you. We have modified the text. 

Comments 5: line 509: health provider91. → health provider [91].
Response 5: thank you. We have modified the text. 

We hope that this revised version of our paper could be suitable for publication

Thank you again for you suggestions

Santilli and co-authors

Reviewer 2 Report

Comments and Suggestions for Authors

This is a review of maternal vaccination and recommendations using current vaccines. The strength of this manuscript is the overview of vaccination of mothers against various infectious agents, and the discussion of how these may affect resistance to infection in the neonate.

However, some of the discussion of immunity is misleading. Specifically, the statements about the transfer of lymphocytes to the infant are not supported by the references given. In addition, there are animal studies indicating that lymphocytes can be transferred to newborns, but I could not find any references that showed that this occurs in humans. The studies referenced showed that anti-specific cells can be found breast milk, but no demonstration of transfer to the baby. Thus, it is recommended that the authors clarify these statements which should include acknowledging limitations. Alternatively, statements about cell mediated immunity being transferred could be deleted, as this would not negatively impact the manuscript.

  1. Lines 74-76: "The fetal and newborn immune system is shifted towards a T-helper 2 (Th2) immune response, thus being ineffective in the response against polysaccharides (i.e. bacterial capsular antigens) and in the defense against intracellular pathogens2." This statement should be reworded. The responses to polysaccharides is a T-independent response, unless linked to a protein (conjugate vaccine), and it is my understanding that infants do not respond well to polysaccharide vaccines is due to the delayed development of a B cell subset, B1. It is not affected by Th2 cell responses. However, because an infant is prone to produce Th2 responses, then the second portion of the sentence, infants being defected in producing a "defense against intracellular pathogens" makes sense. This sentence should rephrased to clarify the points being made.
  2. Lines 204-206: formatting error. It appears that a paragraph return was inadvertently placed in line 204.

Author Response

Dear Reviewer,

We thank you for your positive comments on our paper and for your precious suggestions. We have modified our paper accordingly.

Comments 1 : 

his is a review of maternal vaccination and recommendations using current vaccines. The strength of this manuscript is the overview of vaccination of mothers against various infectious agents, and the discussion of how these may affect resistance to infection in the neonate. However, some of the discussion of immunity is misleading. Specifically, the statements about the transfer of lymphocytes to the infant are not supported by the references given. In addition, there are animal studies indicating that lymphocytes can be transferred to newborns, but I could not find any references that showed that this occurs in humans. The studies referenced showed that anti-specific cells can be found breast milk, but no demonstration of transfer to the baby. Thus, it is recommended that the authors clarify these statements which should include acknowledging limitations. Alternatively, statements about cell mediated immunity being transferred could be deleted, as this would not negatively impact the manuscript.

Response1:  We thank the Reviewer for this comment. The statements were corrected, and literature was updated accordingly. Specifically, placental transfer is not directly mentioned, but a discussion (with appropriate references) about the presence of maternal microchimerism (involving also lymphocytes) is now available.  See lines 135-144 of the tracked version.

Comments 2 :  Lines 74-76: "The fetal and newborn immune system is shifted towards a T-helper 2 (Th2) immune response, thus being ineffective in the response against polysaccharides (i.e. bacterial capsular antigens) and in the defense against intracellular pathogens2." This statement should be reworded. The responses to polysaccharides is a T-independent response, unless linked to a protein (conjugate vaccine), and it is my understanding that infants do not respond well to polysaccharide vaccines is due to the delayed development of a B cell subset, B1. It is not affected by Th2 cell responses. However, because an infant is prone to produce Th2 responses, then the second portion of the sentence, infants being defected in producing a "defense against intracellular pathogens" makes sense. This sentence should rephrased to clarify the points being made.

Response2:   The statement was amended, as correctly suggested. See lines 74-80 of the tracked version. 

Comments 3: Lines 204-206: formatting error. It appears that a paragraph return was inadvertently placed in line 204.

Response 3: Thank you. We have modified the text. 

We sincerely thank you again for your time and consideration, and we hope that the revised manuscript meets your expectations.

Santilli and co-authors

Reviewer 3 Report

Comments and Suggestions for Authors

Introduction: 

Maternal immunization is a key strategy for protecting pregnant individuals and newborns from infectious diseases.  This review examines the mechanisms and benefits of maternal immunization, with a focus on transplacental IgG transfer and immune system interactions.  The authors provide an overview of current recommendations and the safety and efficacy profiles of maternal vaccines, including influenza, tetanus-diphtheria-acellular pertussis (Tdap), respiratory syncytial virus (RSV), COVID-19, and hepatitis B. Additionally, we analyze the barriers to maternal immunization, such as misinformation, vaccine hesitancy, and disparities in healthcare access, while exploring potential strategies to overcome these challenges through targeted educational initiatives, improved provider communication, and policy-driven interventions aimed at increasing vaccine confidence and accessibility.   This review highlights recent innovations and future directions in maternal immunization, including emerging vaccines for Group B Streptococcus and cytomegalovirus.  

Comments:

  1. I would include cautionary language regarding the safety to the child of maternal immunizations, especially with new technologies such as the mRNA COVID-19 vaccines. We have less than five years of data regarding the outcome of vaccination during pregnancy.  There are certainly many examples of delayed adverse reactions to medications given during pregnancy.  Few of the studies cited regarding relatively new vaccines have true longitudinal outcomes five, ten or 20 years down the line. 
  2. Longevity of maternal antibodies in the blood: “However, it has been demonstrated that, after maternal vaccinations, specific antibodies can persist over 6 months.”  Assuming an average half-life of four weeks, the antigen-specific titer of various antibodies should go, by month, 50% of the level at birth, 25%, 12.5%, 6.3%, 3.1%, and 1.6% by six weeks.  There is a reason why young infants typically start suffering with URI at five to six months after birth.
  3. Depending on the vaccine, there is a variable likelihood of transfer of antigen into the fetus. There is no mention of the antigen hierarchy of immune responses in the infant.  There is also no mention of original antigenic sin.  The ability of the fetus to two years of age to mount an antibody response to a variety of antigens is programmed into the system.  Exposure to certain vaccines, e.g. live influenza, has the potential to affect immune responses to that pathogen for years or even decades in the future. 
  4. Concerns “regarding the safety of vaccines for both the mother and the fetus and doubts about vaccine efficacy.” Pregnancy is not a risk-free state.  Between 3 and 4% of births present with congenital or genetic complications that will require medical attention.  Many young mothers who have the unfortunate ill luck for their child to have a clinical issue at birth will seek an explanation, even when there is none.  Many physicians in the US have given in to the temptation to blame vaccination as an inciting event, leading to years of litigation through the Vaccine Injury Compensation Program.  There should be some mention of the potential consequences of physicians or other health care workers inappropriately referring to a vaccination as a cause of child injury

Author Response

Dear Reviewer,

We are grateful for your valuable comments and suggestions, which have helped us to improve the manuscript further.

Comments 1: 
I would include cautionary language regarding the safety to the child of maternal immunizations, especially with new technologies such as the mRNA COVID-19 vaccines. We have less than five years of data regarding the outcome of vaccination during pregnancy.  There are certainly many examples of delayed adverse reactions to medications given during pregnancy.  Few of the studies cited regarding relatively new vaccines have true longitudinal outcomes five, ten or 20 years down the line.

Response 1: 

We appreciate the reviewer’s thoughtful comment and fully agree that maternal immunization, particularly with relatively novel technologies such as mRNA vaccines, warrants ongoing and rigorous safety monitoring. In response, we have revised the manuscript to include a more cautious tone regarding long-term safety data. Specifically, we have added a specific sentence in the text (See lines 324-329 of the tracked version). 

Comments 2: Longevity of maternal antibodies in the blood: “However, it has been demonstrated that, after maternal vaccinations, specific antibodies can persist over 6 months.”  Assuming an average half-life of four weeks, the antigen-specific titer of various antibodies should go, by month, 50% of the level at birth, 25%, 12.5%, 6.3%, 3.1%, and 1.6% by six weeks.  There is a reason why young infants typically start suffering with URI at five to six months after birth.

Response 2: in the revised version, it is now specified that this is a relevant aspect and further studies on this subject are needed. See lines 117-119 of the tracked version

Comments 3:  Depending on the vaccine, there is a variable likelihood of transfer of antigen into the fetus. There is no mention of the antigen hierarchy of immune responses in the infant.  There is also no mention of original antigenic sin.  The ability of the fetus to two years of age to mount an antibody response to a variety of antigens is programmed into the system.  Exposure to certain vaccines, e.g. live influenza, has the potential to affect immune responses to that pathogen for years or even decades in the future. 

Response 3: We thank the Reviewer for this comment. A brief statement on the hierarchy of the immune response in infants was provided (See lines 135 -140 of the tracked version). Additionally, the implications of the original antigenic sin were discussed ( See lines 135-140 of the tracked version) and appropriate references were added.

Comments 4:  Concerns “regarding the safety of vaccines for both the mother and the fetus and doubts about vaccine efficacy.” Pregnancy is not a risk-free state.  Between 3 and 4% of births present with congenital or genetic complications that will require medical attention.  Many young mothers who have the unfortunate ill luck for their child to have a clinical issue at birth will seek an explanation, even when there is none.  Many physicians in the US have given in to the temptation to blame vaccination as an inciting event, leading to years of litigation through the Vaccine Injury Compensation Program.  There should be some mention of the potential consequences of physicians or other health care workers inappropriately referring to a vaccination as a cause of child injury. 

Response 4: We thank the reviewer for this insightful and important comment. We agree that attributing causality between vaccination and congenital or neonatal conditions requires great caution, especially given the emotional and psychological context of childbirth and the natural human inclination to seek explanations for adverse outcomes. As noted, pregnancy is not a risk-free state, and a baseline rate of congenital or genetic complications exists independent of any intervention. In line with this we have added a specific sentence in the text (See line 490-496 in the text).

 We hope that the revisions we have made satisfactorily address the concerns raised.

Round 2

Reviewer 2 Report

Comments and Suggestions for Authors

The authors have appropriately responded to my comments.